# An open label trial of anakinra to prevent respiratory failure in COVID-19

Evdoxia Kyriazopoulou[1], Periklis Panagopoulos[2], Symeon Metallidis[3], George N Dalekos[4], Garyphallia Poulakou[5], Nikolaos Gatselis[4], Eleni Karakike[1], Maria Saridaki[1], Georgia Loli[3], Aggelos Stefos[4], Danai Prasianaki[1], Sarah Georgiadou[4], Olga Tsachouridou[3], Vasileios Petrakis[2], Konstantinos Tsiakos[5], Maria Kosmidou[6], Vassiliki Lygoura[4], Maria Dareioti[1], Haralampos Milionis[6], Ilias C Papanikolaou[7], Karolina Akinosoglou[8], Dimitra-Melia Myrodia[5], Areti Gravvani[5], Aliki Stamou[1], Theologia Gkavogianni[1], Konstantina Katrini[1], Theodoros Marantos[1], Ioannis P Trontzas[5], Konstantinos Syrigos[5], Loukas Chatzis[1], Stamatios Chatzis[1], Nikolaos Vechlidis[1], Christina Avgoustou[1], Stamatios Chalvatzis[1], Miltiades Kyprianou[1], Jos WM van der Meer[9], Jesper Eugen-Olsen[10], Mihai G Netea[9,11], Evangelos J Giamarellos-Bourboulis[1]*

[1]4th Department of Internal Medicine, National and Kapodistrian University of Athens, Athens, Greece; [2]2nd Department of Internal Medicine, Democritus University of Thrace, Medical School, Alexandroupolis, Greece; [3]1st Department of Internal Medicine, Aristotle University ofThessaloniki, Thessaloniki, Greece; [4]Department of Medicine and Research Laboratory of Internal Medicine, National Expertise Center of Greece in Autoimmune Liver Diseases, General University Hospital of Larissa, Larissa, Greece; [5]3rd Department of Internal Medicine, National and Kapodistrian University of Athens, Athens, Greece; [6]1st Department of Internal Medicine,University of Ioannina, School of HealthSciences, Faculty of Medicine, Ioannina, Greece; [7]Department of Pulmonary Medicine, General Hospital of Kerkyra, Kerkyra, Greece; [8]Department of Internal Medicine, University of Patras, Medical School, Rion, Greece; [9]Department of Internal Medicine and Center for Infectious Diseases, Radboud University, Nijmegen, Netherlands; [10]Clinical Research Centre, Copenhagen University Hospital Hvidovre, Hvidovre, Denmark; [11]Immunology and Metabolism, Life & Medical Sciences Institute, University of Bonn, Bonn, Germany

*For correspondence:
egiamarel@med.uoa.gr

## Abstract

**Background:** It was studied if early suPAR-guided anakinra treatment can prevent severe respiratory failure (SRF) of COVID-19.

**Methods:** A total of 130 patients with suPAR ≥6 ng/ml were assigned to subcutaneous anakinra 100 mg once daily for 10 days. Primary outcome was SRF incidence by day 14 defined as any respiratory ratio below 150 mmHg necessitating mechanical or non-invasive ventilation. Main secondary outcomes were 30-day mortality and inflammatory mediators; 28-day WHO-CPS was explored. Propensity-matched standard-of care comparators were studied.

**Results:** 22.3% with anakinra treatment and 59.2% comparators (hazard ratio, 0.30; 95% CI, 0.20–0.46) progressed into SRF; 30-day mortality was 11.5% and 22.3% respectively (hazard ratio 0.49; 95% CI 0.25–0.97). Anakinra was associated with decrease in circulating interleukin (IL)−6, sCD163 and sIL2-R; IL-10/IL-6 ratio on day 7 was inversely associated with SOFA score; patients were allocated to less severe WHO-CPS strata.

**Conclusions:** Early suPAR-guided anakinra decreased SRF and restored the pro-/anti-inflammatory balance.

**Funding:** This study was funded by the Hellenic Institute for the Study of Sepsis, Technomar Shipping Inc, Swedish Orphan Biovitrum, and the Horizon 2020 Framework Programme.
**Clinical trial number:** NCT04357366.

## Introduction

Patients with severe infections caused by the novel coronavirus SARS-CoV-2 (known as COVID-19) have high circulating concentrations of pro-inflammatory cytokines. Although these concentrations are often not as high as in patients with classical acute respiratory distress syndrome (ARDS) and septic shock (*Sinha et al., 2020*), systemic inflammation is an important feature of severe COVID-19. It has been shown that when severe respiratory failure (SRF) necessitating mechanical ventilation (MV) emerges, two separate immune phenomena predominate in the infected host; (i) macrophage activation syndrome involving 25% patients; or (ii) complex immune dysregulation with down-regulation of the human leukocyte antigen DR on circulating monocytes, lymphopenia, and over-production of interleukin (IL)−6 by monocytes involving 75% of the patients (*Giamarellos-Bourboulis et al., 2020*).

It is hypothesized that these immune reactions start early in patients with lower respiratory tract infection (LRTI) by SARS-CoV-2 and are progressively enhanced so as to lead to SRF. This has been suggested to be due to the early release of IL-1 from the lung epithelial cells that are infected by the virus; IL-1 stimulates further cytokine production from alveolar macrophages (*Conti et al., 2020*).

As a consequence, it is assumed that early start of anti-IL-1 anti-inflammatory treatment may prevent SRF (*van de Veerdonk and Netea, 2020*). However, most probably not all patients with LRTI by SARS-CoV-2 are in need of early treatment and a screening tool to capture those who are likely to progress to SRF would be an asset. Soluble urokinase plasminogen activator receptor (suPAR) seems to be such a screening tool. We and others recently demonstrated that suPAR concentrations above 6 ng/ml herald worsening to SRF 14 days earlier (*Rovina et al., 2020*; *Azam et al., 2020*). The positive predictive value for the early prediction of SRF was as high as 85.9%. uPAR is anchored to the cell membranes of the lung endothelial cells. As a result of the activation of kallikrein, uPAR is cleaved and enters the systemic circulation as the soluble counterpart suPAR (*Pixley et al., 2011*).

We conducted the SAVE trial (suPAR-guided Anakinra treatment for Validation of the risk and Early management of SRF by COVID-19) to investigate whether the early administration of anakinra in patients with LRTI due to SARS-CoV-2 and suPAR equal to or greater than 6 µg/l may prevent the development of SRF. Anakinra is the recombinant soluble IL-1 receptor antagonist that blocks both IL-1α and IL-1β. We report herein the results of the interim analysis in the first 130 enrolled patients and compare the efficacy of anakinra with parallel patients receiving standard-of-care (SOC) treatment.

## Results

### Trial conduct

Interim analysis was performed when the 30-day follow-up of the first 130 patients was completed. The first patient was enrolled on April 16, 2020 and the last on September 12, 2020. Registration at the EU repository EudraCT was done on March 31, 2020 before submission for regulatory approval. Although the study is on-going, we present here the results of the pre-planned interim analysis described in the amendment of the SAVE trial on October 15, 2020 to the National Organization for Medicines of Greece. The inclusion of 179 parallel SOC comparators was done within the same time frame. The baseline values of Acute Physiology and Chronic Health Evaluation (APACHE) II score, Sequential Organ Failure Assessment (SOFA) score, pneumonia severity index (PSI), white blood cells, and ferritin of all SOC comparators were different from the 130 participants in the SAVE trial. To match for these differences, propensity score-matching was done and 130 fully matched parallel SOC comparators were selected (*Table 1*). The study flow chart is shown in *Appendix 1—figure 1*. In all, 211 patients were excluded because they had suPAR less than 6 ng/ml; patients were followed-up and SRF developed in seven (3.3%) patients. Baseline demographics of patients receiving

**eLife digest** People infected with the SARS-CoV-2 virus, which causes COVID-19, can develop severe respiratory failure and require a ventilator to keep breathing, but this does not happen to every infected individual. Measuring a blood protein called suPAR (soluble urokinase plasminogen activator receptor) may help identify patients at the greatest risk of developing severe respiratory failure and requiring a ventilator. Previous investigations have suggested that measuring suPAR can identify pneumonia patients at highest risk for developing respiratory failure. The protein can be measured by taking a blood sample, and its levels provide a snapshot of how the body's immune system is reacting to infection, and of how it may respond to treatment.

Anakinra is a drug that forms part of a class of medications called interleukin antagonists. It is commonly prescribed alone or in combination with other medications to reduce pain and swelling associated with rheumatoid arthritis. Kyriazopoulou et al. investigated whether treating COVID-19 patients who had developed pneumonia with anakinra could prevent the use of a ventilator and lower the risk of death.

The findings show that treating COVID-19 patients with an injection of 100 milligrams of anakinra for ten days may be an effective approach because the drug combats inflammation. Kyriazopoulou et al. examined various markers of the immune response and discovered that anakinra was able to improve immune function, protecting a significant number of patients from going on a ventilator. The drug was also found to be safe and cause no significant adverse side effects.

Administering anakinra decreased of the risk of progression into severe respiratory failure by 70%, and reduced death rates significantly. These results suggest that it may be beneficial to use suPAR as an early biomarker for identifying those individuals at highest risk for severe respiratory failure, and then treat them with anakinra. While the findings are promising, they must be validated in larger studies.

---

anakinra with SOC treatment and of patients receiving only SOC treatment are shown in *Table 1*; baseline demographics did not differ. The level of care offered in the two groups was also similar (*Appendix 1—table 1*).

## Study endpoints

Twenty-nine patients (22.3%; 95% CI, 16.0–30.2%) among the intent-to-treat (ITT) population receiving anakinra and SOC progressed to SRF until day 14. The incidence of SRF among the 130 parallel SOC-treated comparators was 59.2% (n = 77) (95% CI, 50.6–67.3%, p<0.0001) (hazard ratio 0.30; 95% CI, 0.20–0.46) (*Figure 1A* and *Table 2*). The superiority of anakinra treatment was also documented over the total of 179 available SOC parallel comparators (*Figure 1—figure supplement 1*). However, the baseline differences between the total SOC parallel comparators and the participants of the SAVE trial led to limit the remaining analysis between the 130 anakinra-treated participants of the SAVE trial and the 130 fully matched SOC parallel comparators. Multivariate step-wise Cox regression analysis for variables showed that anakinra treatment was the only independent variable protective from SRF (hazard ratio 0.28; 95% CI, 0.18–0.44; p<0.0001) (*Table 3*). The reported higher frequency of dexamethasone intake among patients who developed SRF should not be interpreted as causality; it does simply reflect that the prescription of dexamethasone was greater among patients who were considered more severe by the treating physicians. One separate multivariate step-wise Cox regression analysis among patients treated with dexamethasone showed anakinra to be the only independent variable protective from SRF (*Appendix 1—table 2*).

Anakinra treatment was of benefit in most of the secondary study endpoints, that is, 30-day mortality; absolute change of SOFA score by day 14; and absolute change of the respiratory symptoms score by days 7 and 14 (*Table 2*). Mortality of participants in the SAVE trial receiving anakinra treatment after 30 days was 11.5% (95% CI, 7.1–18.2%); this was 22.3% (95% CI, 16.0–30.2%) in parallel comparators receiving SOC (*Figure 1B*). Multivariate step-wise Cox regression analysis for variables showed that anakinra treatment was the only independent variable protective from 30-day mortality (hazard ratio 0.49; 95% CI 0.25–0.97; p: 0.041) (*Appendix 1—table 3*).

**Table 1.** Baseline characteristics of patients.

| | SOC + anakinra (N = 130) | Parallel SOC (N = 179) | p-value* | Parallel SOC after propensity matching (N = 130) | p-value** |
|---|---|---|---|---|---|
| Age, years, mean (SD) | 63 (14) | 66 (14) | 0.094 | 64 (14) | 0.839 |
| Male sex, No. (%) | 81 (62.3) | 116 (64.8) | 0.719 | 84 (64.6) | 0.797 |
| Days from onset of symptoms to start of treatment, median (range) | 8 (1–23) | 6 (4–9) | 0.012 | 7 (1–12) | 0.143 |
| Severity indexes, mean (SD) | | | | | |
| Charlson's comorbidity index | 3 (2) | 3 (2) | 0.123 | 3 (2) | 0.927 |
| APACHE II score | 7 (3) | 8 (5) | 0.002 | 7 (3) | 0.957 |
| SOFA score | 2 (1) | 3 (2) | 0.001 | 2 (1) | 0.813 |
| Pneumonia severity index | 69 (22) | 80 (28) | <0.0001 | 70 (20) | 0.709 |
| WHO classification for COVID-19, No. (%) | | | | | |
| Moderatepneumonia, no oxygen | 61 (46.9) | 68 (38.0) | 0.158 | 57 (43.8) | 0.709 |
| Severe pneumonia | 69 (53.1) | 111 (62.0) | | 73 (56.2) | |
| Comorbidities, No. (%) | | | | | |
| Type 2 diabetes mellitus | 41 (31.5) | 45 (25.1) | 0.248 | 32 (24.6) | 0.270 |
| Chronic heart failure | 11 (8.5) | 19 (10.6) | 0.565 | 10 (7.7) | 1.000 |
| Chronic renal disease | 1 (0.8) | 12 (6.7) | 0.009 | 4 (3.1) | 0.370 |
| Coronary heart disease | 10 (7.7) | 24 (13.4) | 0.141 | 14 (10.8) | 0.521 |
| Arterial hypertension | 68 (52.3) | 87 (48.6) | 0.565 | 57 (43.8) | 0.214 |
| Chronic obstructive pulmonary disease | 12 (9.2) | 14 (7.8) | 0.682 | 6 (4.6) | 0.221 |
| Solid tumor malignancy | 8 (6.2) | 14 (7.8) | 0.658 | 9 (6.9) | 1.000 |
| Cerebrovascular disease | 7 (5.4) | 13 (7.3) | 0.641 | 4 (3.1) | 0.540 |
| Bacterial co-infection, No. (%) | 1 (0.8) | 5 (2.8) | 0.407 | 3 (2.3) | 0.622 |
| *Streptococcus pneumoniae* | 0 (0.0) | 2 (1.1) | 0.511 | 2 (1.5) | 0.498 |
| *Escherichia coli* | 0 (0.0) | 1 (0.6) | 1.000 | 1 (0.8) | 1.000 |
| Laboratory values, median ($Q_1$–$Q_3$) | | | | | |
| White blood cells, cells/mm$^3$ | 5400 (4390–6830) | 6215 (4805–9013) | 0.005 | 5879 (4760–7800) | 0.106 |
| Lymphocytes, cells/mm$^3$ | 950 (722–1252) | 961 (615–1271) | 0.307 | 977 (587–1318) | 0.382 |
| Platelets, cells/mm$^3$ | 184,000 (137,000–246,000) | 202,000 (150,000–270,000) | 0.121 | 204,400 (150,450–271,625) | 0.107 |
| C-reactive protein, mg/l | 47.4 (14.3–105.5) | 64.7 (18.3–140.0) | 0.179 | 68.8 (19.7–141.8) | 0.117 |
| Procalcitonin, ng/ml | 0.14 (0.08–0.31) | 0.23 (0.10–0.71) | 0.149 | 0.13 (0.08–0.41) | 0.841 |
| Ferritin, ng/ml | 536.5 (280.0–898.5) | 629.5 (375.0–1277.0) | <0.001 | 607.5 (367.8–1196.0) | 0.107 |
| Serum soluble uPAR, ng/ml | 8.9 (7.0–12.2) | 9.2 (7.0–13.8) | 0.344 | 9.0 (7.0–11.7) | 0.973 |
| PaO$_2$/FiO$_2$, mmHg | 293.3 (195.7–371.2) | 262.0 (182.0–350.3) | 0.090 | 285.7 (208.5–371.7) | 0.917 |
| Concomitant treatment, No. (%) | | | | | |
| Hospitalization at tertiary academic hospitals | 127 (97.7) | 156 (87.2) | 0.191 | 120 (92.3) | 0.167 |
| β-lactamase inhibitors | 18 (13.8) | 16 (8.9) | 0.199 | 13 (10) | 0.444 |
| Third generation cephalosporins | 60 (46.2) | 66 (36.9) | 0.127 | 43 (33.1) | 0.042 |
| Piperacillin/tazobactam | 37 (28.7) | 61 (34.1) | 0.323 | 42 (32.3) | 0.590 |
| Ceftaroline | 38 (29.2) | 50 (27.9) | 0.800 | 42 (32.3) | 0.687 |
| Carbapenem | 11 (8.5) | 23 (12.8) | 0.271 | 15 (11.5) | 0.536 |
| Moxifloxacin/levofloxacin | 27 (20.9) | 26 (14.5) | 0.170 | 17 (13.2) | 0.136 |
| Glycopeptides | 2 (1.5) | 6 (3.4) | 0.475 | 5 (3.8) | 0.447 |

*Table 1 continued on next page*

Table 1 continued

| | SOC + anakinra (N = 130) | Parallel SOC (N = 179) | p-value* | Parallel SOC after propensity matching (N = 130) | p-value** |
|---|---|---|---|---|---|
| Azithromycin | 96 (73.8) | 145 (81.0) | 0.164 | 104 (80.0) | 0.303 |
| Remdesivir | 8 (6.2) | 12 (6.7) | 1.000 | 11 (8.5) | 0.635 |
| Hydroxychloroquine | 56 (43.1) | 92 (51.4) | 0.066 | 68 (52.3) | 0.172 |
| Dexamethasone | 52 (40.0) | 65 (36.3) | 0.553 | 47 (36.2) | 0.610 |
| Predicted probability | 0.488 (0.386–0.571) | _ | _ | 0.488 (0.391–0.556) | 0.629 |

Abbreviations: APACHE: acute physiology and chronic health evaluation; n: number; Q: quartile; SD: standard deviation; SOC: standard-of-care; SOFA: sequential organ failure assessment; uPAR: urokinase-type plasminogen activator receptor; WHO: World Health Organization.

*p-value of comparison between SOC + anakinra treated patients and parallel standard-care comparators.

**p-value of comparison between SOC + anakinra treated patients and parallel standard-care comparators after propensity matching.

Two main secondary study outcomes were the effects of anakinra treatment on circulating inflammatory biomarkers and on the function of peripheral blood mononuclear cells (PBMCs). Compared to SOC comparators, anakinra-treated subjects experienced increase of the absolute lymphocyte count and decreases of IL-6, sCD163, and sIL-2R (*Figure 2*). The IL-10/IL-6 ratio of serum (an index of the anti-inflammatory/pro-inflammatory balance in severe COVID-19 [*McElvaney et al., 2020*]) was inversely associated with the absolute increase of the SOFA score on day 14 among anakinra-treated patients, compatible with the anti-inflammatory effect of anakinra. Remarkably, suPAR was increased among anakinra-treated patients on day 7 from baseline. Thus, anakinra is associated with protection against progression to SRF even when the suPAR signal indicates an unfavorable outcome. The function of PBMCs of patients was modulated among anakinra-treated patients. More precisely, the production of IL-1β and IL-10 on day 7 was greater among patients who did not develop SRF pointing toward a restoration of the ability of the PBMCs to adapt to balanced production of anti-inflammatory and pro-inflammatory cytokines. This was further corroborated with the positive association between the IL-10/IL-1β ratio of production from PBMCs on day 7 with the ratio of serum IL-10/IL-6 of the same day (*Figure 3*).

The exploratory endpoints were ventilator-free days until day 28, the 28-day World Health Organization clinical progression scale (WHO-CPS), 90-day mortality, and the cost of hospitalization. The number of ventilator-free days until day 28 was increased with anakinra treatment (*Table 2*) and 90-day mortality was decreased (*Table 2* and *Appendix 1—figure 2*). Participants of the SAVE trial were allocated to strata of lower severity by day 28 compared to propensity-matched parallel SOC comparators (*Figure 4*). The overall cost of hospitalization decreased from median €2398.4 among SOC treated comparators to €1291.4 among anakinra-treated patients (*Appendix 1—figure 3*).

## Safety

The adverse events (AEs) and serious adverse events (SAEs) that were captured during the study period of 14 days are listed in *Table 4*. The incidence of the same events was depicted among SOC treated comparators. As shown in *Table 4*, the incidence of these events was not greater in the anakinra group than comparators, with the only exception of leukopenia having a trend to be higher in the anakinra group. SAEs were fewer among anakinra-treated patients.

## Discussion

Anakinra treatment of COVID-19 patients admitted with LRTI and suPAR concentrations greater or equal than 6 µg/l was associated with a relative decrease of the incidence of SRF by 70%. Anakinra-treated patients who were eventually admitted to the ICU had a shorter stay than those who did not receive anakinra. So apparently the benefit of previous anakinra treatment remained. This is also reflected by the overall decrease of mortality by day 30 and by day 90. An important point of the SAVE study is the strategy to select for early anakinra treatment using the predictive biomarker suPAR. In previous studies in COVID-19 and in sepsis, this marker turned out to be able to predict

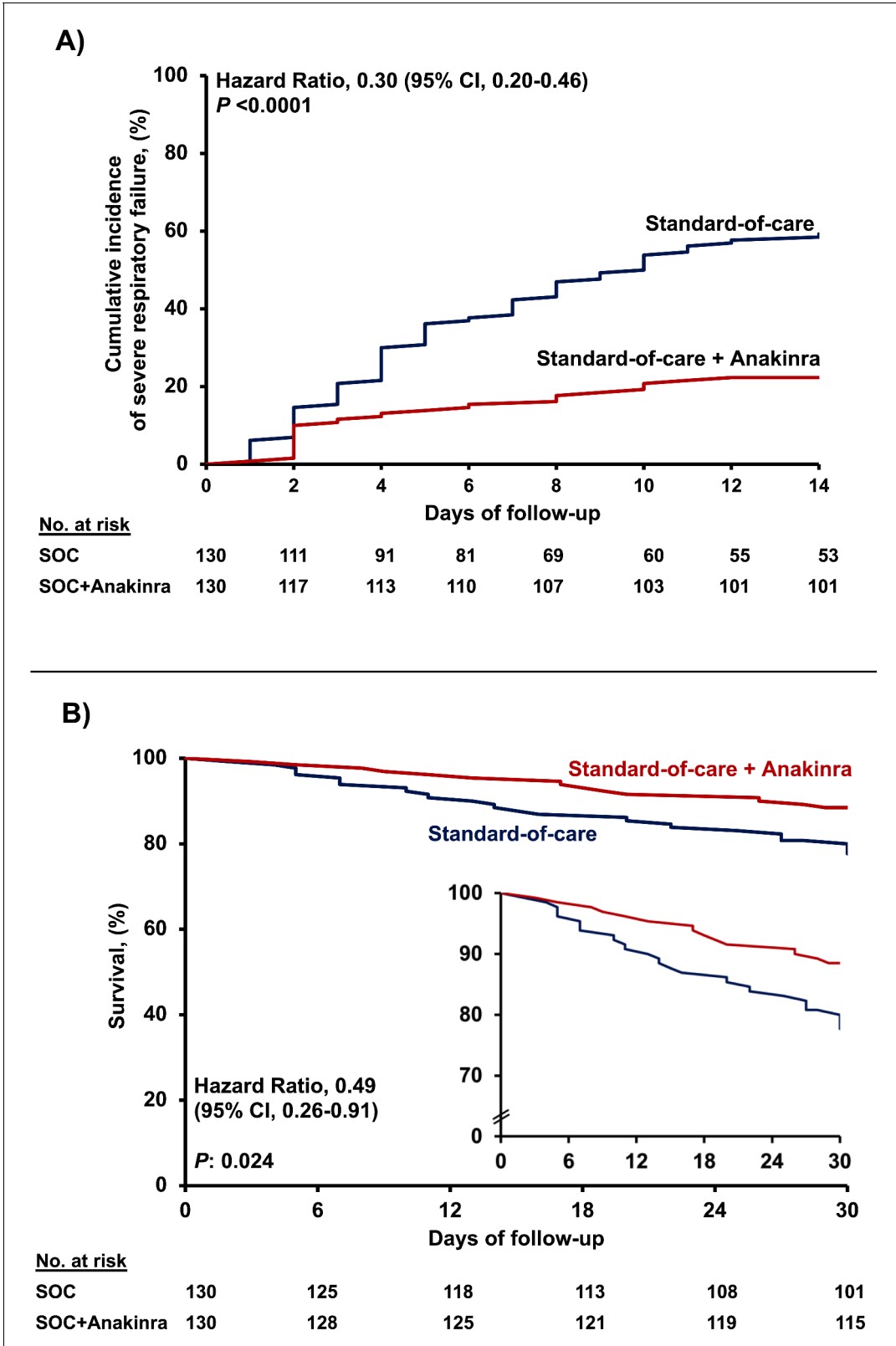

**Figure 1.** Outcome of patients treated with anakinra. (**A**) Curves of the cumulative incidence of severe respiratory failure (SRF); and (**B**) curves of 30-day mortality; the inset shows the curves in enlarged Y-axis. The analysis includes propensity-scored fully matched comparators treated with standard-of-care (SOC; n = 130) and the 130 participants in the SAVE trial who were treated with SOC treatment and anakinra (n = 130). Separate analysis for the
*Figure 1 continued on next page*

*Figure 1 continued*

cumulative incidence of SRF between the total available 179 comparators and the 130 participants in the SAVE trial can be found in *Figure 1—figure supplement 1*.

The online version of this article includes the following figure supplement(s) for figure 1:

**Figure supplement 1.** Incidence of SRF among the total of 179 available comparators treated with SOC before matching and the 130 partiicpants in the SAVE trial treated with Anakinra and SOC.

the likelihood for unfavorable outcome (*Hayek et al., 2020*; *Rovina et al., 2020*; *Azam et al., 2020*; *Giamarellos-Bourboulis et al., 2012*).

Other studies have reported favorable effects of anakinra treatment in COVID-19 pneumonia. In a retrospective analysis by Cavalli et al., 29 patients with respiratory failure and respiratory ratio below 100 mmHg were treated with high dose anakinra intravenously (5 mg/kg twice daily). Treatment was associated with clinical improvement in 72% and remarkably higher survival rate (*Cavalli et al., 2020*). In the Ana-COVID prospective study, 52 patients with confirmed COVID-19 and bilateral lung infiltrates and oxygen saturation less than 93% were treated with 100 mg anakinra subcutaneously twice daily for 3 days followed by 100 mg subcutaneously once daily for another 7 days. The study had a composite endpoint much similar to the endpoint of the SAVE study; that is, MV and/or death. Anakinra treatment achieved a 78% decrease of this composite endpoint compared to 44 historical controls. However, studied comparators were not matched for co-administered medication such as azithromycin and hydroxychloroquine (*Huet et al., 2020*). In a retrospective study, 12 patients with COVID pneumonia and increased C-reactive protein were intravenously treated with anakinra 300 mg/day; none died (*Cauchois et al., 2020*).

It is difficult to compare these three studies as enrolled patients had different stages of COVID and variable disease severity. Hence the clinical dilemma is which patients would benefit from

**Table 2.** Primary, secondary, and exploratory study outcomes.

| | Parallel SOC (N = 130) | SOC + Anakinra after propensity matching (N = 130) | OR (95% CI) | p-value |
|---|---|---|---|---|
| Severe respiratory failure by day 14, No. (%) | 77 (59.2) | 29 (22.3) | 0.19 (0.12–0.34) | **<0.0001** |
| Mechanical ventilation (MV) by day 14, No. (%) | 65 (50.0) | 25 (19.2) | 0.24 (0.14–0.42) | **<0.0001** |
| Non-invasive MV by day 14, No. (%) | 9 (6.9) | 6 (4.6) | 0.65 (0.23–1.88) | 0.596 |
| 14-day mortality, No. (%) | 16 (12.3) | 6 (4.6) | 0.35 (0.13–0.91) | **0.043** |
| SOFA score on day 7, median ($Q_1$–$Q_3$) | 2 (0 to 3) | 1 (0 to 3) | NA | 0.540 |
| SOFA score on day 14, median ($Q_1$–$Q_3$) | 2 (0–9) | 1 (0–3) | NA | 0.191 |
| Absolute change of SOFA score by day 7 compared to baseline, median ($Q_1$–$Q_3$) | 0 (−1 to 1) | 0 (−1 to 0) | NA | 0.356 |
| Absolute change of SOFA score by day 14 compared to baseline, median ($Q_1$–$Q_3$) | 0 (−1 to 6) | −2 (−1 to 0) | NA | **0.004** |
| Absolute change of respiratory symptoms score by day 7 compared to baseline, median ($Q_1$–$Q_3$) | 0 (0 to 0.75) | −1 (−3 to 0) | NA | **0.019** |
| Absolute change of respiratory symptoms score by day 14 compared to baseline, median ($Q_1$–$Q_3$) | 0 (−0.75 to 0) | −2 (−4 to −1) | NA | **0.016** |
| Ventilator-free days, mean (SD) | 18 (11) | 25 (8) | NA | **<0.0001** |
| 30-day mortality, No. (%) | 29 (22.3) | 15 (11.5) | 0.45 (0.23–0.90) | **0.031** |
| 90-day mortality, No. (%) | 40 (30.8) | 22 (16.9) | 0.46 (0.25–0.83) | **0.013** |

Abbreviations: CI: confidence interval; n: number; NA: non-applicable; OR: odds ratio; Q: quartile; SD: standard deviation; SOC; standard-of-care; SOFA: sequential organ failure assessment.

**Table 3.** Anakinra as an independent protective factor from development of severe respiratory failure (SRF) by day 14. Univariate and multivariate (Cox forward conditional) models, for patients receiving anakinra with standard-of-care (SOC) treatment and for parallel comparators receiving only SOC treatment are presented. Only admission variables that differ significantly between patients who developed and those who did not develop SRF by day 14 are provided. Results are provided after four steps of analysis.

| Variable, no. (%) | SRF (−) (N = 154) | SRF (+) (N = 106) | Univariate analysis | | Multivariate analysis | |
|---|---|---|---|---|---|---|
| | | | HR (95% CI) | p-value | HR (95% CI) | p-value |
| Anakinra treatment, n (%) | 101 (65.6) | 29 (27.4) | 0.30 (0.20–0.47) | <0.0001 | 0.28 (0.18–0.44) | <0.0001 |
| APACHE II, mean (SD) | 6 (3) | 8 (3) | 1.12 (1.07–1.19) | <0.0001 | | |
| SOFA score, mean (SD) | 2 (1) | 3 (1) | 1.58 (1.38–1.82) | <0.0001 | 1.41 (1.21–1.65) | <0.0001 |
| Pneumonia severity index, mean (SD) | 66 (22) | 73 (19) | 1.01 (1.00–1.02) | 0.017 | | |
| Severe COVID-19 by WHO classification, n (%) | 65 (42.2) | 77 (72.6) | 2.78 (1.81–4.27) | <0.0001 | 1.74 (1.09–2.79) | 0.020 |
| Soluble uPAR, ng/ml, median ($Q_1$–$Q_3$) | 8.3 (6.7–11.0) | 10.0 (8.0–13.9) | 1.10 (1.04–1.17) | 0.001 | 1.07 (1.01–1.14) | 0.022 |
| Lymphocytes/mm$^3$, median ($Q_1$–$Q_3$) | 1000 (733–1315) | 930 (579–1197) | 0.99 (0.99–1.00) | 0.026 | | |
| C-reactive protein, mg/l, median ($Q_1$–$Q_3$) | 41.1 (12.7–95.2) | 73.4 (30.6–153.1) | 1.00 (1.00–1.00) | <0.0001 | | |
| Ferritin, ng/ml, median ($Q_1$–$Q_3$) | 525.0 (280.0–804.5) | 641.5 (451.3–1556.8) | 1.00 (1.00–1.00) | <0.0001 | | |
| PaO$_2$/FiO$_2$, mmHg, median ($Q_1$–$Q_3$) | 330.8 (231.9–386.7) | 244.4 (161.7–305.8) | 0.99 (0.99–0.99) | <0.0001 | | |
| Treatment with third generation cephalosporin, n (%) | 73 (47.4) | 30 (28.3) | 0.53 (0.35–0.82) | 0.004 | | |
| Treatment with piperacillin/tazobactam, n (%) | 37 (24.0) | 42 (39.6) | 1.60 (1.08–2.38) | 0.019 | | |
| Treatment with carbapenem, n (%) | 7 (4.5) | 19 (17.9) | 2.79 (1.69–4.60) | <0.0001 | 2.19 (1.26–3.83) | 0.006 |
| Treatment with glycopeptide, n (%) | 1 (0.6) | 6 (5.7) | 2.60 (1.14–5.95) | 0.023 | | |
| Treatment with dexamethasone, n (%) | 46 (29.9) | 53 (50.0) | 1.73 (1.19–2.54) | 0.005 | | |

Abbreviations: APACHE: acute physiology and chronic health evaluation; CI: confidence interval; HR: hazard ratio; n: number; PaO$_2$/FiO$_2$: ratio of partial oxygen pressure to the fraction of inspired oxygen; Q: quartile; SD: standard deviation; SOFA: sequential organ failure assessment; SRF: severe respiratory failure; uPAR: urokinase-type plasminogen activator receptor.

anakinra, in what stage of COVID, and in what dosage regimen. In this respect, the SAVE study gives guidance: the best candidates are patients with high likelihood of SRF as defined by increased suPAR; anakinra presented one well-acceptable safety profile at the standard subcutaneous daily dose of 100 mg.

An important additional finding of the SAVE trial is that it provides mechanistic insight into the biological effects of anakinra: the treatment was associated with a reset of the pro- versus anti-inflammatory balance of the host. The production capacity of PMBCs for the anti-inflammatory IL-10 was increased and this was reflected by the serum IL-10/IL-6 ratio. IL-10/IL-6 is an index of the anti-inflammatory/pro-inflammatory balance which is severely disturbed in severe COVID-19, much more than in bacterial sepsis (*McElvaney et al., 2020*). The described reset in the production capacity of the PBMCs was linked to clinical benefit since the serum IL-10/IL-6 ratio was inversely associated with the absolute change of SOFA score. Anakinra treatment also decreased the elevated serum concentrations of sCD163 and sIL2-R that are biomarkers of macrophage activation (*Bleesing et al., 2007*; *Rubin, 1990*). It was noted before that patients with COVID-19 who deteriorate have characteristics of macrophage activation (*Zhou et al., 2020*); the decrease of these biomarkers indicates attenuation of macrophage activation among anakinra-treated patients.

Recent data suggest early activation of the NLRP3 inflammasome in severe COVID-19; the formation of the end product caspase-1 was enhanced among patients who had unfavorable outcome. Stimulation of human monocytes with SARS-CoV-2 could not induce production of IL-1β and priming was needed. Findings suggest that over-produced IL-1β is derived from the cleavage of pro-IL-1β induced by SARS-CoV-2 (*Rodrigues et al., 2021*). While anakinra treatment is meant to inhibit IL-1 bioactivity through blocking of its receptor, it may also inhibit the early production of IL-1β following NLRP3 activation due to the autocrine effects of IL-1 (*Dinarello et al., 1987*). Anakinra also inhibits IL-1α. SARS-CoV-2 infection is suggested to cause massive release of IL-1α followed by sensing from monocytes and tissue macrophages and further activation of the NLRP3 inflammasome leading to perpetuation of the pro-inflammatory responses (*Cavalli et al., 2021*).

The lack of randomized design is acknowledged as a limitation in study design. The non-randomized design led to two more limitations: the use of SOC parallel comparators and the lack of availability of follow-up samplings on day 7. The SAVE trial was designed in mid-March 2020 at the beginning of the pandemic in Greece. It was chosen to adapt an open-label and single-arm design in an attempt to help as many patients as possible since no SOC was framed at that time period. The SOC parallel comparators were optimally matched since matching was based on similarities in timeframe of enrolment, level of care, baseline severity, and co-administered treatment. When the study was started, dexamethasone treatment was not yet part of the SOC treatment but in the following months many patients received dexamethasone. Since dexamethasone acts as a cytokine-inhibiting agent, it is an important question how strong would the effect of anakinra be when patients also receive dexamethasone. Although this is not a preset endpoint, post hoc analysis shows that anakinra is also protective in patients receiving dexamethasone (*Appendix 1—table 2*). Several meetings held between the investigators addressed the need to adapt one placebo comparator arm of treatment; however, the available data about the efficacy of anakinra did not lead to take the decision of integration of placebo comparators.

This interim analysis was presented to the Emergency Task Force for COVID-19 of the European Medicine Agency. Provided advice led to the design of the pivotal, confirmatory phase III randomized clinical trial with the acronym SAVE-MORE (EudraCT number: 2020-005828-11; Clinicaltrial.gov NCT04680949) which is actually running in 40 study sites; 32 sites in Greece and eight sites in Italy.

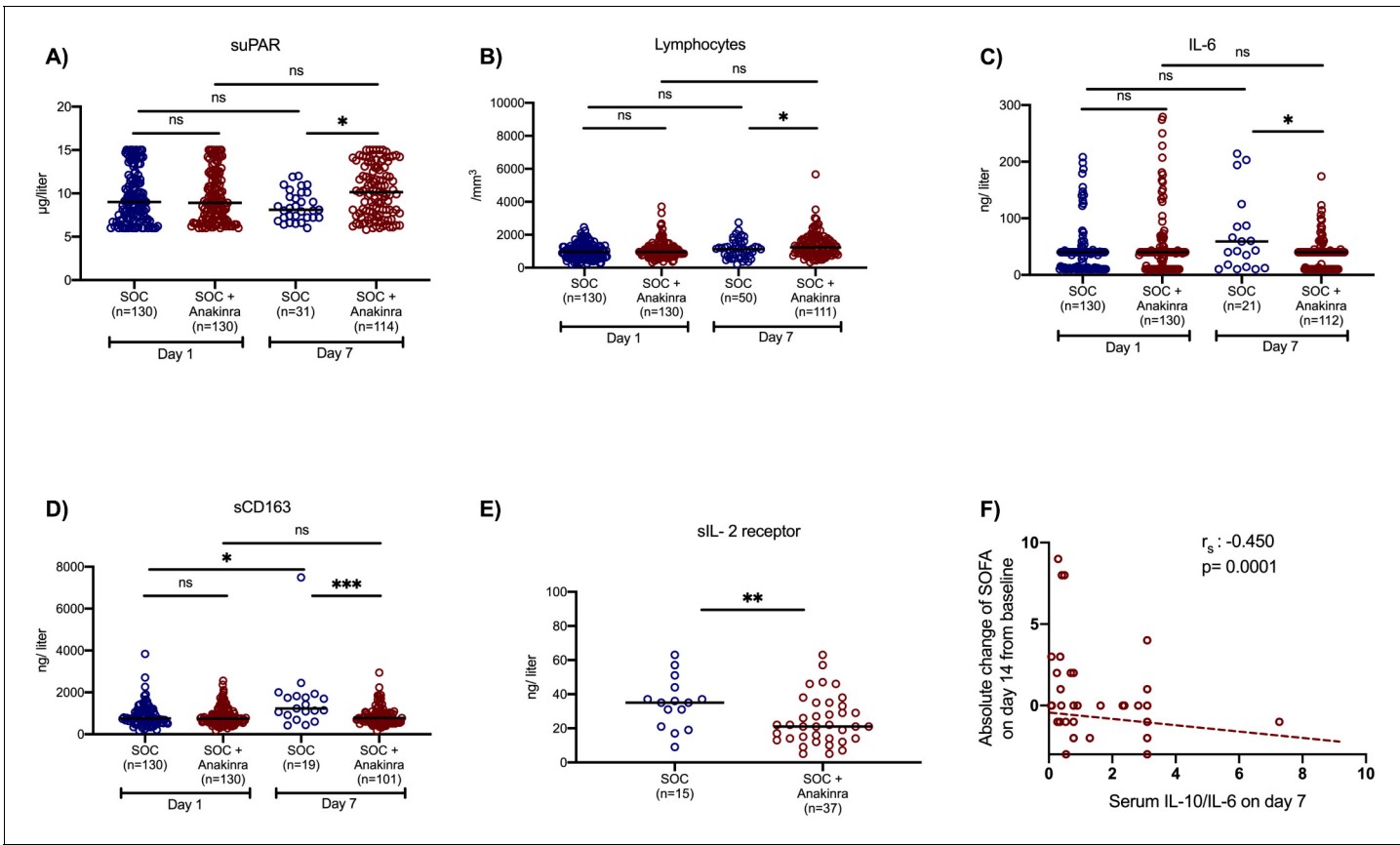

**Figure 2.** Effect of anakinra treatment on circulating mediators of patients with lower respiratory tract infection by SARS-CoV-2. Concentrations of (A) suPAR (soluble urokinase plasminogen activator receptor); (B) the absolute lymphocyte count; (C) interleukin (IL)-6; and (D) sCD163 of patients under parallel standard-of-care (SOC) treatment (blue color) and of patients under treatment with SOC and anakinra (red color) at baseline (day of hospital admission) and day 7 of follow-up are shown. The concentrations of sIL-2R on day 7 are provided in panel E. Lines refer to median values. Comparisons between different groups were performed by the Mann–Whitney U-test and within the same group by the Wilcoxon test. Statistical comparisons: ns, non-significant; *p<0.05; **p<0.01; ***p<0.0001. The correlation of the IL-10/IL-6 ratio with the absolute change of the SOFA (sequential organ failure assessment) score on day 14 from the baseline is also shown (panel F). The Spearman rank of order correlation co-efficient and the p-value of the correlation is provided.

In conclusion, we propose a novel strategy using suPAR as an early biomarker that can effectively identify those patients at high risk for SRF. In these patients, prophylactic treatment with regular doses of anakinra is associated with prevention of the incidence of SRF. The restoration of the pro-inflammatory/anti-inflammatory balance is proposed as the mechanism of anakinra action.

# Materials and methods

## Key resources table

| Reagent type (species) or resource | Designation | Source or reference | Identifiers | Additional information |
|---|---|---|---|---|
| Commercial assay or kit | Human IL-1b uncoated ELISA | Invitrogen | 88–7261 | |
| Commercial assay or kit | Human IL-6 uncoated ELISA | Invitrogen | 88–7066 | |
| Commercial assay or kit | Human IL-10 uncoated ELISA | Invitrogen | 88–7066 | |
| Commercial assay or kit | Human IL-2R uncoated ELISA | Invitrogen | 88-7025-22 | |
| Commercial assay or kit | Human sCD163 uncoated ELISA | Affymetrix Inc | RAB0082 | |
| Commercial assay or kit | Human Ferritin ELISA | ORGENTEC Diagnostika GmbH | ORG 5FE | |
| Chemical compound, drug | Lymphosep, Lymphocyte Separation Media | Biowest | L0560 | |
| Chemical compound, drug | PBS Dulbecco's Phosphate Buffered Saline w/o Magnesium, w/o Calcium | Biowest | L0615 | |
| Chemical compound, drug | FBS Superior; standardized Fetal Bovine Serum, EU-approved | Biochrom | S0615 | |
| Chemical compound, drug | Gentamycin Sulfate BioChemica | PanReac AppliChem | A1492 | |
| Chemical compound, drug | Penicillin G Potassium Salt BioChemica | PanReac AppliChem | A1837 | |
| Chemical compound, drug | Lipopolysaccharides from *Escherichia coli* O55:B5 | Sigma-Aldrich | L2880 | |

## Trial oversight

SAVE is an ongoing open-label non-randomized trial conducted in six study sites in Greece (EudraCT number 2020-001466-11; National Ethics Committee approval 38/20; National Organization for Medicines approval ISO 28/20; ClinicalTrials.gov registration NCT04357366). Parallel comparators receiving SOC treatment were hospitalized at the same time period in seven departments of Internal Medicine in tertiary hospitals of Greece who were participating in the registry of the Hellenic Sepsis Study Group (HSSG) without participating in the SAVE trial (*Appendix 1—table 1*) (http://www.sep-sis.gr). All consecutive admissions in the six study sites where the SAVE trial was conducted and in the seven study sites where parallel SOC comparators were hospitalized were screened for eligibility. The trial was conducted by the Hellenic Institute for the Study of Sepsis (HISS) and funded in part by HISS, by Technomar Shipping Inc, by Swedish Orphan Biovitrum AB and by the Horizon 2020 RISKinCOVID grant. The funders had no role in the design, conduct, analysis and interpretation of data, and decision to publish. The laboratory of Immunology of Infectious Diseases of the 4th

Department of Internal Medicine at ATTIKON University General Hospital served as a central laboratory for the study. The initial draft of the manuscript was written by the first and the last authors. All authors vouch for the adherence of the trial to the protocol and first and last authors vouch for the accuracy and completeness of the data and analysis.

## Patients

Enrolled patients were adults hospitalized with confirmed infection by SARS-CoV-2 virus by real-time PCR reaction of nasopharyngeal secretions; radiological findings compatible with LRTI; and plasma suPAR level ≥6 μg/l using the suPARnostic Quick Triage kit (Virogates S/A, Blokken 45, 3460 Birkerød, Denmark). Exclusion criteria were: any stage IV malignancy; any do not resuscitate decision; ratio of partial oxygen pressure to the fraction of inspired oxygen ($pO_2/FiO_2$) less than 150 mmHg;

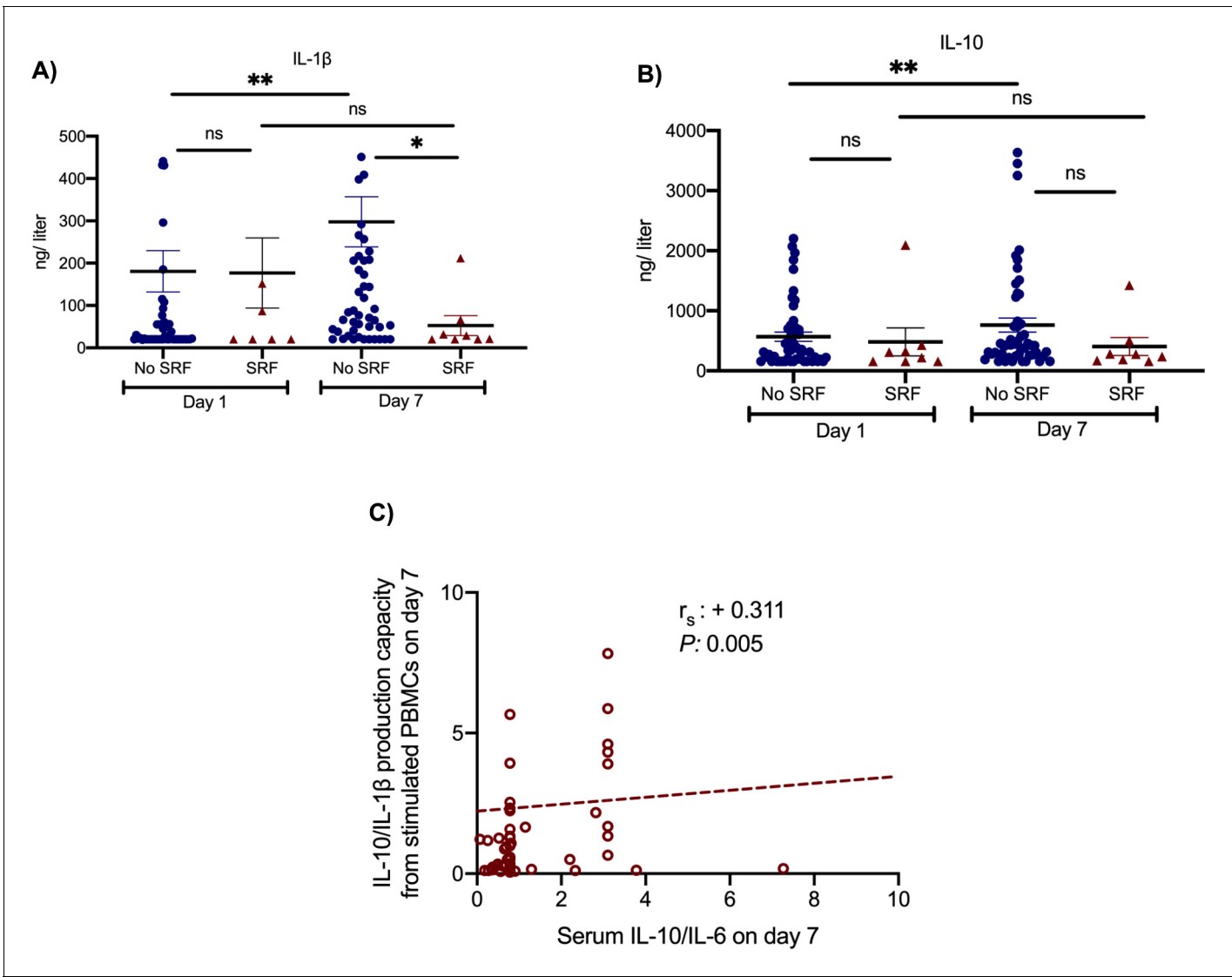

**Figure 3.** Effect of anakinra treatment on cytokine production capacity from peripheral blood mononuclear cells (PBMCs). PBMCs were isolated before and 7 days after start of treatment with anakinra. PBMCs were stimulated with lipopolysaccharide (LPS) of *Escherichia coli* O55:B5 for the production of interleukin (IL)−1β and with heat-killed *Candida albicans* for the production of IL-10. Production of IL-1β (panel **A**) and of IL-10 (panel **B**) is presented separately for patients who developed by day 14 severe respiratory failure (SRF) or not. Lines refer to median values. Comparisons between different groups were performed by the Mann–Whitney U-test and within the same group by the Wilcoxon test. Statistical comparisons: ns, non-significant; *p<0.05; **p<0.01. The correlation between the ratio of IL-10/IL-1β production by PBMCs and of the ratio of IL-10/IL-6 in serum on day 7 is provided in panel **C**. The Spearman rank of order correlation co-efficient and the p-value of the correlation is provided.

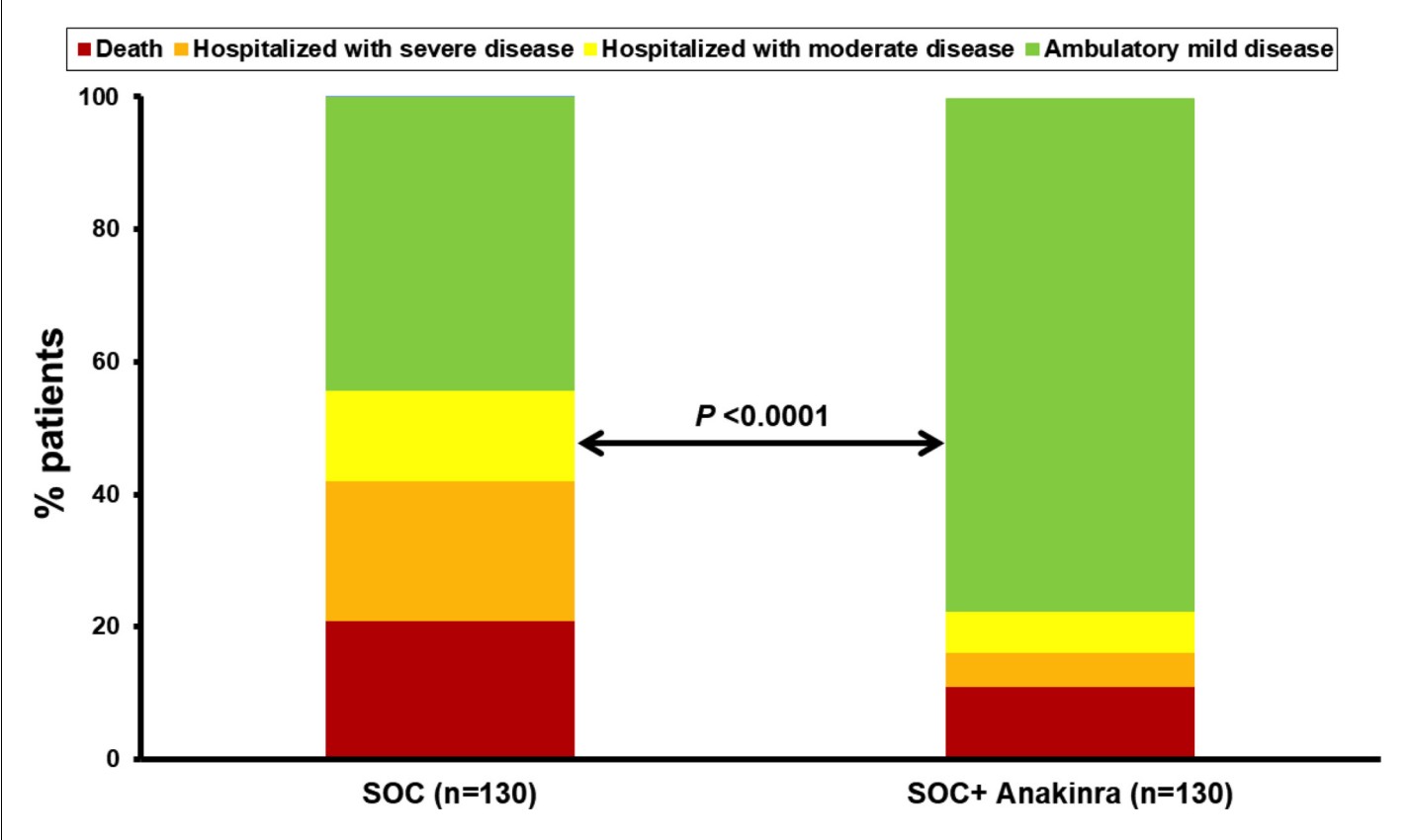

**Figure 4.** Allocation of patients into the strata of the WHO clinical progression scale by day 28. The analysis includes propensity-scored fully matched comparators treated with standard-of-care (SOC, n = 130) and the 130 participants in the SAVE trial who were treated with SOC treatment and anakinra (n = 130). Comparisons were done by the Pearson's chi-square test.

need for MV or non-invasive ventilation under positive pressure (NIV); any primary immunodeficiency; neutropenia (<1500/mm$^3$); any intake of corticosteroids at a daily dose $\geq$0.4 mg/kg prednisone or equivalent the last 15 days; any anti-cytokine biological treatment the last month; and pregnancy or lactation. The procedure to identify appropriate comparators treated in parallel with SOC was as follows: (a) patients receiving SOC were hospitalized in seven medical departments that participate in the HSSG (http://www.sepsis.gr). These departments are active in the research network of the HSSG since 2006 where they collaborate with the six medical departments participating in the SAVE trial. They are tertiary departments providing to severe patients SOC treatment regularly updated with the current guidelines; (b) the period of hospitalization was exactly the same as the period of hospitalization of the participants in the SAVE trial; (c) selection required that patients were meeting all the inclusion criteria of the SAVE trial; and (d) selection required that patients did not meet any of the exclusion criteria of the SAVE trial. Clinical data of these patients on the day of hospital admission and start of SOC were used for comparison to the baseline data of participants of the SAVE trial. Written informed consent was provided by the patient or legal representative before screening.

## Trial interventions

Enrolled patients received 100 mg anakinra subcutaneously once daily for 10 days. All other drugs were allowed. Fifteen milliliters of whole blood was collected before start and 7 days after start of anakinra and collected into EDTA-coated tubes and sterile and pyrogen-free tubes for the isolation of PBMCs, serum, and plasma. PBMCs were stimulated for cytokine production. Blood was sampled from parallel SOC comparators on the day of hospital admission and repeated after 7 days; plasma

**Table 4.** Adverse events and serious adverse events.

The incidence of the same events was depicted among parallel SOC treated, propensity-matched comparators.

| | Parallel SOC (N = 130) | SOC + anakinra (N = 130) | OR (95% CI) | p-value |
|---|---|---|---|---|
| At least one SAE by day 14, No. (%) | 63 (48.5) | 32 (24.6) | 0.34 (0.21–0.59) | <0.0001 |
| Extended hospitalization | 63 (48.5) | 32 (24.6) | 0.34 (0.21–0.59) | <0.0001 |
| Death | 16 (12.3) | 6 (4.6) | 0.35 (0.13–0.91) | 0.043 |
| Shock | 56 (43.1) | 27 (20.8) | 0.34 (0.20–0.59) | <0.0001 |
| Acute kidney injury | 37 (28.5) | 15 (11.5) | 0.32 (0.17–0.63) | 0.001 |
| Any bacterial infection | 30 (23.1) | 9 (6.9) | 0.22 (0.10–0.49) | <0.0001 |
| Thromboembolic event | 5 (3.8) | 2 (1.5) | 0.32 (0.06–1.72) | 0.188 |
| Pulmonary edema | 0 (0) | 1 (0) | NA | 1.00 |
| At least one AE by day 14, No. (%) | 89 (68.5) | 85 (65.4) | 0.87 (0.51–1.45) | 0.693 |
| Gastrointestinal disturbances | 9 (6.9) | 15 (11.5) | 1.74 (0.74–4.17) | 0.284 |
| Electrolyte abnormalities | 41 (31.5) | 35 (26.9) | 0.80 (0.47–1.37) | 0.496 |
| Elevated liver function tests | 51 (39.2) | 40 (30.8) | 0.69 (0.41–1.15) | 0.193 |
| Anemia | 26 (20.0) | 22 (16.9) | 0.82 (0.44–1.53) | 0.632 |
| Leukopenia | 3 (2.3) | 11 (8.5) | 3.91 (1.07–14.37) | 0.051 |
| Thrombopenia | 7 (5.4) | 9 (6.9) | 1.31 (0.47–3.62) | 0.797 |
| Headache | 2 (1.5) | 4 (3.1) | 2.03 (0.37–11.29) | 0.684 |
| Allergic reaction | 7 (5.4) | 4 (3.1) | 0.56 (0.16–1.95) | 0.540 |
| Any heart arrhythmia | 22 (16.9) | 9 (6.9) | 0.37 (0.16–0.83) | 0.020 |

Abbreviations: AE: non-serious adverse event; CI: confidence interval; n: number; NA: non-available; OR: odds ratio; SAE: serious adverse event; SOC: standard-of-care.

and serum were prepared. Biomarkers and cytokines were measured in plasma, serum, and supernatants of PBMC cultures.

PBMCs were isolated after gradient centrifugation over Ficoll (Biochrom, Berlin, Germany) for 20 min at 1400 g. After three washings in ice-cold PBS pH 7.2, PBMCs were counted in a Neubauer plate with trypan blue exclusion of dead cells. They were then diluted in RPMI 1640 enriched with 2 mM of L-glutamine, 500 µg/ml of gentamicin, 100 U/ml of penicillin G, 10 mM of pyruvate, 10% fetal bovine serum (Biochrom), and suspended in wells of a 96-well plate. The final volume per well was 200 µl with a density of $2 \times 10^6$ cells/ml. PBMCs were exposed in duplicate for 24 hr or 5 days at 37°C in 5% $CO_2$ to different stimuli: 10 ng/ml of *Escherichia coli* O55:B5 lipopolysaccharide (LPS, Sigma, St. Louis, USA) or $5 \times 10^5$ colony forming units of heat-killed *Candida albicans*. F Concentrations of IL-1β, IL-6, and IL-10 were measured in cell supernatants or serum in duplicate by an enzyme immunoassay (Invitrogen, Carlsbad, California, USA). The lowest detections limits were: for IL-1β 10 pg/ml; for IL-6 10 pg/ml; and for IL-10 5 pg/ml. Concentrations of ferritin (ORGENTEC Diagnostika GmbH, Mainz, Germany), sCD163 (Affymetrix Inc, Santa Clara, CA), and sIL-2R (ORGENTEC Diagnostika GmbH, Mainz, Germany) were measured in serum by an enzyme-immunoassay; the lower limit of detection was 75 ng/ml for ferritin; 0.31 ng/ml for sCD163; and 0.5 ng/ml for sIL-2R.

Hospitalization cost was calculated per patient in Euros as the sum of all administered medicines and the addition of the nominal cost of daily stay in the intensive care unit or in the general ward. The unit price for counted items derived from the official pricelist as defined by the Greek government (KYA 4$^α$/οικ.13740/27.03.2012; government gajette 4898 τB/01.1.2018). The cost of human resources (salaries of nursing and medical personnel) was not counted.

## Outcomes

The incidence of SRF by day 14 was the primary outcome. SRF was defined as any decrease of $pO_2$/$FiO_2$ below 150 necessitating MV or NIV (*Giamarellos-Bourboulis et al., 2020*). Patients dying before day 14 were considered meeting the primary endpoint. Secondary outcomes were 30-day

mortality; and the changes of respiratory symptoms score (*Ramirez et al., 2019*; *Stets et al., 2019*; *Barrera et al., 2016*), of SOFA score, of PBMC cytokine stimulation and of circulating plasma inflammatory mediators between days 1 and 7. Ventilator-free days by day 28, the 28-day WHO-CPS, 90-day mortality, and the cost of hospitalization were exploratory endpoints.

AEs (Common Terminology Criteria for Adverse Events, version 4.03) and SAEs were captured.

## Statistical analysis

The sample size was calculated assuming the incidence of SRF would decrease from 60% to 45% with anakinra treatment. To achieve so with 90% power at the 5% level of significance, 260 patients were needed. An interim analysis was planned when the first 130 patients would be enrolled. Analysis of the primary endpoint was done by the ITT principle.

Qualitative data were presented as percentages with confidence intervals (CI) and quantitative data as median with quartiles. Among all parallel comparators receiving SOC treatment, 130 comparators were selected by propensity score 1:1 matching with patients treated with anakinra with SOC. Matching criteria were: age; Charlson's comorbidity index (CCI); admission severity scores namely PSI, APACHE II score, SOFA score, and WHO severity classification of COVID-19; and SOC treatment with azithromycin, hydroxychloroquine, and dexamethasone. Comparison with anakinra-treated patients was performed by the Fisher's exact test using confirmatory forward stepwise Cox analysis (IBM SPSS Statistics v. 25.0). Comparisons of cytokines and mediators between groups were done by the Student's 't-test' for parametric variables; and by the Mann–Whitney U-test for non-parametric variables. Paired comparisons were done by the Wilcoxon's rank-signed test. Non-parametric correlations were done according to Spearman. Cost comparisons were done by the Mann–Whitney U-test. Any two-sided p-value < 0.05 was statistically significant (please refer also to *Supplementary file 1*: Protocol and statistical analysis plan).

## Acknowledgements

The authors would like to thank the patients, families, and clinical, laboratory, and research staff who contributed to the trial. The study was supported in part by the Hellenic Institute for the Study of Sepsis, in part by Technomar Shipping Inc, in part by Swedish Orphan Biovitrum AB, and in part by the Horizon 2020 grant RISKinCOVID.

## Additional information

### Competing interests

Evangelos J Giamarellos-Bourboulis: Reviewing editor, *eLife*. Jos WM van der Meer: Senior editor, *eLife*. Periklis Panagopoulos: honoraria from GILEAD Sciences, Janssen, and MSD. George N Dalekos: Advisor/Lecturer for Abbvie, Bristol-Myers Squibb, Gilead, Novartis, Roche, Amgen, MSD, Janssen, Ipsen and Pfizer, has received Grant support from Bristol-Myers Squib, Gilead, Roche, Janssen, Abbvie and Bayer and was or is currently PI in National & International Protocols sponsored by Abbvie, Bristol-Myers Squibb, Novartis, Gilead, Novo Nordisk, Genkyotex, Regulus Therapeutics Inc, Tiziana Life Sciences, Bayer, Astellas, Ipsen, Pfizer and Roche. Garyphallia Poulakou: independent educational grants from Pfizer, MSD, Angelini, and Biorad. Haralampos Milionis: honoraria, consulting fees and non-financial support from healthcare companies, including Amgen, Angelini, Bayer, Mylan, MSD, Pfizer, and Servier. Jesper Eugen-Olsen: cofounder, shareholder and CSO of ViroGates A7S, Denmark and is named inventor on patents on suPAR owned by Copenhagen University Hospital Hvidovre, Denmark. Mihai G Netea: supported by an ERC Advanced Grant (#833247) and a Spinoza grant of the Netherlands Organization for Scientific Research. He has also received independent educational grants from TTxD, GSK and ViiV HealthCare. The other authors declare that no competing interests exist.

### Funding

| Funder | Grant reference number | Author |
|---|---|---|
| Hellenic Institute for the Study | | Evangelos J Giamarellos- |

| of Sepsis | | Bourboulis |
| Technomar Shipping Inc | | Evangelos J Giamarellos-Bourboulis |
| Swedish Orphan Biovitrum | | Evangelos J Giamarellos-Bourboulis |
| Horizon 2020 Framework Programme | RISKinCOVID | Evangelos J Giamarellos-Bourboulis |

The funders had no role in study design, data collection and interpretation, or the decision to submit the work for publication.

### Author contributions
Evdoxia Kyriazopoulou, Formal analysis, Writing - original draft, Project administration; Periklis Panagopoulos, Symeon Metallidis, George N Dalekos, Garyphallia Poulakou, Nikolaos Gatselis, Eleni Karakike, Maria Saridaki, Georgia Loli, Aggelos Stefos, Danai Prasianaki, Sarah Georgiadou, Olga Tsachouridou, Vasileios Petrakis, Konstantinos Tsiakos, Maria Kosmidou, Vassiliki Lygoura, Maria Dareioti, Haralampos Milionis, Ilias C Papanikolaou, Karolina Akinosoglou, Dimitra-Melia Myrodia, Areti Gravvani, Aliki Stamou, Theologia Gkavogianni, Konstantina Katrini, Theodoros Marantos, Ioannis P Trontzas, Konstantinos Syrigos, Loukas Chatzis, Stamatios Chatzis, Nikolaos Vechlidis, Christina Avgoustou, Stamatios Chalvatzis, Investigation, Writing - review and editing; Miltiades Kyprianou, Formal analysis, Writing - review and editing; Jos WM van der Meer, Jesper Eugen-Olsen, Writing - review and editing; Mihai G Netea, Writing - original draft; Evangelos J Giamarellos-Bourboulis, Conceptualization, Funding acquisition, Project administration, Writing - review and editing

### Author ORCIDs
Evdoxia Kyriazopoulou (iD) http://orcid.org/0000-0002-9585-517X
Maria Kosmidou (iD) https://orcid.org/0000-0002-8618-9435
Loukas Chatzis (iD) https://orcid.org/0000-0002-2832-0116
Evangelos J Giamarellos-Bourboulis (iD) https://orcid.org/0000-0003-4713-3911

### Ethics
Clinical trial registration EudraCT number 2020-001466-11 and https://clinicaltrials.gov/ registration NCT04357366.
Human subjects: Approved in Greece by National Ethics Committee approval 38/20; National Organization for Medicines approval ISO 28/20. Written informed consent was provided by the patient or legal representative before screening.

### Decision letter and Author response
Decision letter https://doi.org/10.7554/eLife.66125.sa1
Author response https://doi.org/10.7554/eLife.66125.sa2

## Additional files

### Supplementary files
• Supplementary file 1. Protocol and statistical analysis plan.

• Transparent reporting form

### Data availability
Data of this submission are anticipated to be part of a submission package to the European Medicines Agency for the request of approval of Anakinra for the management of COVID-19 guided by the biomarker suPAR. Once this is finalized, the data will be made publicly available upon request. Requests will require signing contract with the sponsor of the study which is the Hellenic Institute for the Study of Sepsis. The responsible official is Ms Leda Efstratiou who is the DPO responsible for

GDPR. Interested researchers should contact Ms Efstratiou at http://www.lestyre.org/contact.html and headed@sepsis.gr to request access to the data.

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

## Appendix 1

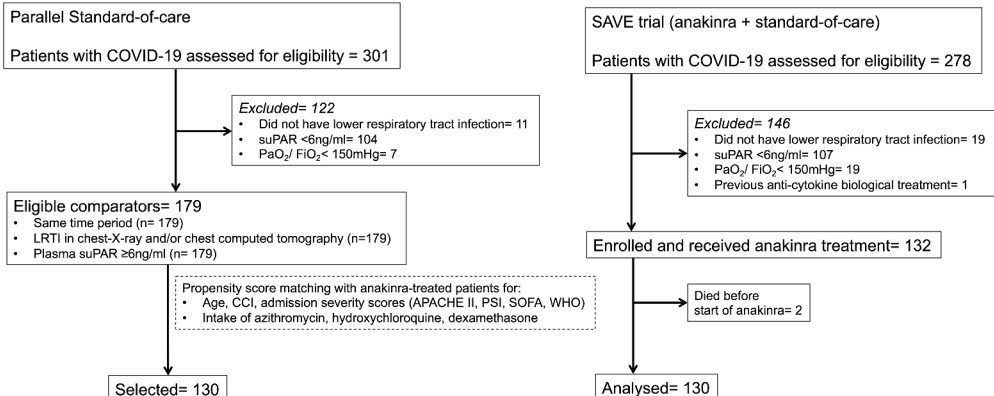

**Appendix 1—figure 1.** Trial profile and selection of parallel standard-of-care (SOC) comparators. SOC comparators were hospitalized at the same time period at seven department of Internal Medicine in tertiary hospitals and they were selected in two steps: at the first step after application of the inclusion and exclusion criteria of patients receiving anakinra; and at the second step after propensity score matching. Abbreviations: APACHE II: acute physiology and chronic health evaluation; CCI: Charlson's comorbidity index; COVID-19: infection by the new coronavirus SARS-CoV-2; ITT: intent-to-treat; LRTI: lower respiratory tract infection; PaO$_2$/FiO$_2$: respiratory fraction of partial oxygen pressure to fraction of inspired oxygen; PSI: pneumonia severity index; SOFA: sequential organ failure assessment; suPAR: soluble urokinase plasminogen activator receptor; WHO: severity classification of COVID-19 by the World Health organization.

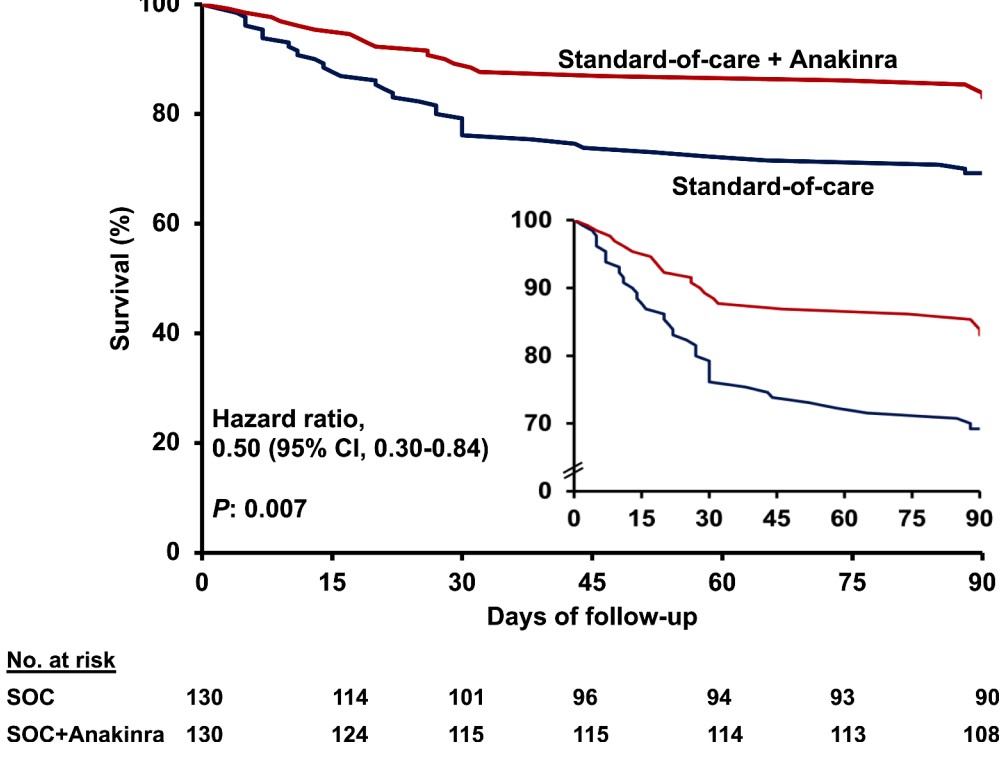

*Appendix 1—figure 2 continued on next page*

**Appendix 1—figure 2.** Survival curves until day 90. Patients participating in the SAVE trial received standard-of-care (SOC) treatment and anakinra and were compared to parallel comparators treated only with SOC. The inset shows the curves in enlarged Y-axis. CI: confidence interval.

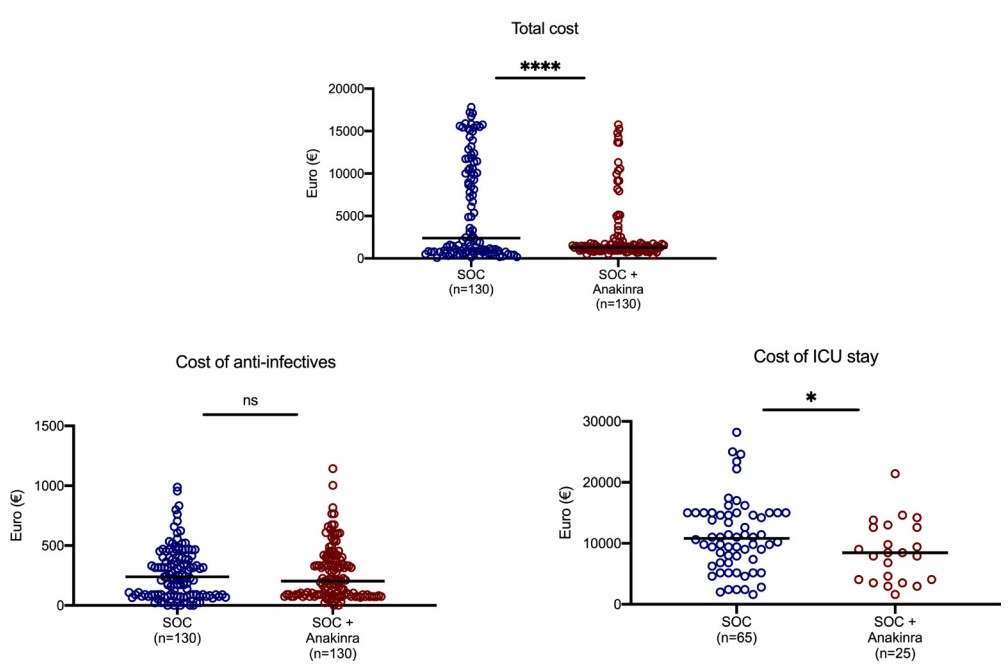

**Appendix 1—figure 3.** Cost of hospitalization. The three main categories of cost are shown: total cost, cost of anti-infectives, and cost of stay in the intensive care unit. Statistical comparisons between groups. ns: non-significant; *p<0.05; ****p<0.0001. Abbreviation: SOC: standard-of-care.

**Appendix 1—table 1.** List of participating study sites where patients were hospitalized.

| Name of study site | City | Level of care | SOC (n) | SOC + anakinra (n) |
|---|---|---|---|---|
| 4th Department of Internal Medicine, ATTIKON University Hospital | Athens | Tertiary Academic | 17 | |
| 3rd Department of Internal Medicine, Sotiria General Hospital | Athens | Tertiary Academic | | 26 |
| 1st Department of Pulmonary Medicine, Sotiria General Hospital | Athens | Tertiary Academic | 25 | |
| AHEPA General Hospital COVID-19 unit I | Thessaloniki | Tertiary Academic | | 43 |
| AHEPA General Hospital COVID-19 unit II | Thessaloniki | Tertiary Academic | 32 | |
| University General Hospital of Larissa | Larissa | Tertiary Academic | | 36 |
| University General Hospital of Alexandroupolis, COVID-19 unit I | Alexandroupolis | Tertiary Academic | | 18 |
| University General Hospital of Alexandroupolis, COVID-19 unit II | Alexandroupolis | Tertiary Academic | 18 | |
| University General Hospital of Ioannina, COVID-19 unit I | Ioannina | Tertiary Academic | | 3 |

*Continued on next page*

*Appendix 1—table 1 continued*

| Name of study site | City | Level of care | SOC (n) | SOC + anakinra (n) |
|---|---|---|---|---|
| University General Hospital of Ioannina, COVID-19 unit II | Ioannina | Tertiary Academic | 11 | |
| University General Hospital of Patras | Patras | Tertiary Academic | 15 | |
| 1st Department of Internal Medicine, Thriasio General Hospital | Athens | Tertiary NHS | 12 | |
| Department of Pulmonary Medicine, Kerkyra General Hospital | Kerkyra | Tertiary NHS | | 4 |

Abbreviations N: number; NHS: national health system; SOC: standard-of-care.

**Appendix 1—table 2.** Anakinra as an independent protective factor from development of severe respiratory failure (SRF) by day 14 among patients treated with dexamethasone.
Univariate and multivariate (Cox forward conditional) models, for the anakinra with standard-of-care group (SOC) and parallel SOC comparators are presented. Only admission variables that differ significantly between patients who developed and those who did not develop SRF by day 14 are provided.

| | | | Univariate analysis | | Multivariate analysis | |
|---|---|---|---|---|---|---|
| Variable no. (%) | SRF (−) (N = 46) | SRF (+) (N = 53) | HR (95% CI) | p-value | HR (95% CI) | p-value |
| Anakinra treatment, n (%) | 31 (67.4) | 21 (39.6) | 0.55 (0.32–0.97) | 0.038 | 0.56 (0.32–0.97) | 0.038 |
| Soluble uPAR, ng/ml, median ($Q_1$–$Q_3$) | 9.60 (7.2–12.6) | 11.6 (7.3–12.6) | 1.10 (1.00–1.20) | 0.030 | 1.97 (1.14–3.41) | 0.018 |
| Ferritin, ng/ml, median ($Q_1$–$Q_3$) | 593.0 (411.3–968.5) | 633.0 (412.0–968.0) | 1.00 (1.00–1.00) | 0.025 | | |

Abbreviations: CI: confidence interval; HR: hazard ratio; n: number; Q: quartile; SRF: severe respiratory failure; uPAR: urokinase-type plasminogen activator receptor.

**Appendix 1—table 3.** Anakinra as an independent protective factor from 30-day mortality.
Univariate and multivariate (Cox forward conditional) models, for the anakinra with standard-of-care (SOC) group and parallel SOC comparators are presented. Only admission variables that differ significantly between patients who died and those who did not die are provided. Results are provided after four steps of analysis.

| | | | Univariate analysis | | Multivariate analysis | |
|---|---|---|---|---|---|---|
| Parameters, no. (%) | Survival (N = 216) | Death (N = 44) | HR (95% CI) | p-value | HR (95% CI) | p-value |
| Anakinra treatment, n(%) | 115 (53.2) | 15 (34.1) | 0.49 (0.26–0.91) | 0.024 | 0.49 (0.25–0.97) | 0.041 |
| Charlson's comorbidity index, mean (SD) | 3 (2) | 4 (2) | 1.29 (1.13–1.48) | <0.0001 | 1.26 (1.09–1.48) | 0.002 |
| APACHE II score, mean (SD) | 6 (3) | 9 (3) | 1.24 (1.14–1.35) | <0.0001 | | |
| SOFA score, mean (SD) | 2 (1) | 3 (1) | 2.13 (1.73–2.63) | <0.0001 | 2.03 (1.61–2.55) | <0.0001 |
| Pneumonia severity index, mean (SD) | 67 (21) | 81 (18) | 3.66 (1.85–7.24) | <0.0001 | | |

*Continued on next page*

*Appendix 1—table 3 continued*

| Parameters, no. (%) | Survival (N = 216) | Death (N = 44) | Univariate analysis HR (95% CI) | p-value | Multivariate analysis HR (95% CI) | p-value |
|---|---|---|---|---|---|---|
| Severe COVID-19 by WHO classification, n (%) | 105 (48.6) | 37 (84.1) | 4.86 (2.17–10.92) | 0.0001 | | |
| Soluble uPAR, ng/ml, median ($Q_1$–$Q_3$) | 8.8 (6.8–11.4) | 10.4 (8.0–14.7) | 1.15 (1.05–1.26) | 0.003 | | |
| Lymphocytes/mm$^3$, median ($Q_1$–$Q_3$) | 984 (699–1275) | 839 (545–1167) | 0.999 (0.99–1.00) | 0.024 | | |
| C-reactive protein, mg/l, median ($Q_1$–$Q_3$) | 50.7 (17.5–118.4) | 80.1 (18.7–156.7) | 1.003 (1.00–1.01) | 0.033 | | |
| $PaO_2$/$FiO_2$ mmHg, median ($Q_1$–$Q_3$) | 301 (224.6–374.4) | 195.8 (88.2–290.4) | 0.99 (0.98–0.99) | <0.0001 | | |
| History of type 2 diabetes mellitus, n (%) | 53 (24.5) | 20 (45.5) | 2.38 (1.31–4.31) | 0.004 | | |
| Treatment with third generation cephalosporin, n (%) | 93 (43.1) | 10 (22.7) | 0.43 (0.21–0.88) | 0.020 | | |
| Treatment with piperacillin/ tazobactam, n (%) | 59 (27.3) | 20 (45.5) | 2.03 (1.11–3.69) | 0.021 | | |
| Treatment with remdesivir, n (%) | 11 (5.1) | 8 (18.6) | 3.22 (1.50–6.95) | 0.003 | 1.27 (1.09–1.47) | 0.002 |

<u>Abbreviations</u>: APACHE: acute physiology and chronic health evaluation; CI: confidence interval; HR: hazard ratio; n: number; $PaO_2$/$FiO_2$: ratio of partial oxygen pressure to the fraction of inspired oxygen; q: quartile; SD: standard deviation; SOFA: sequential organ failure assessment; uPAR: urokinase-type plasminogen activator receptor.

