## [Decision Letter]

**Acceptance summary:**

The authors have used suPAR (soluble urokinase plasminogen activator receptor) levels to identify and select a high-risk group of patients which they argue are more likely to develop severe respiratory failure (SRF). Recruitment was strong with an interim analysis performed 5 months after commencement with 130 patients recruited in the treatment group. As the authors achieved the primary outcome, a reduction in D14 SRF, in the treatment group the results are now reported. Other secondary outcomes including cytokine analysis generally support the hypothesis that antagonism of a hyper-inflammatory state using the IL-1 receptor antagonist anakinra have favorable effects in high-risk COVID-19 patients.

**Decision letter after peer review:**

Thank you for submitting your article "An open label trial of anakinra to prevent respiratory failure in COVID-19" for consideration by *eLife*. Your article has been reviewed by three peer reviewers, and the evaluation has been overseen by a Reviewing Editor and a Senior Editor. The following individual involved in review of your submission has agreed to reveal their identity: Charles Dinarello (Reviewer #3).

The reviewers and the Reviewing Editor have discussed the reviews with one another. This decision letter summarizes the points raised to help you prepare a revised submission.

Summary:

Comparing a prospective, but open label and single treatment, assignment of a recombinant interleukin-1 receptor antagonist (anakinra) in a high-risk population of patients admitted with pneumonia due to COVID-19 with a concurrent, matched group of patients treated without anakinra, the authors report a reduced rate of severe respiratory failure at day 14. The authors have used suPAR (soluble urokinase plasminogen activator receptor) levels to select a high-risk group of patients they argue are more likely to develop severe respiratory failure (SRF). Recruitment was strong with an interim analysis performed 5 months after commencement with 130 patients recruited in the treatment group. As the authors achieved the primary outcome, a reduction in D14 SRF, in the treatment group the results are reported. Other secondary outcomes including cytokine analysis generally support the hypothesis that antagonism of a hyper-inflammatory state would have beneficial effects.

Essential Revisions:

1) Cytokine storm – there is concern particularly from Carolyn Calfee and Michael Matthay that COVID-19 patients with ARDS do not have higher median cytokine levels than other patients with ARDS or septic shock (Sinha, Matthay and Calfee, 2020 and Lancet Resp Med Dec 2020 https://doi.org/10.1016/S2213-2600(20)30404-5). Given that various well-designed double blind trials have not shown benefit from this approach despite likely greater cytokine levels than your cohort I suggest this argument be rewritten and great care used when referring to a cytokine storm.

2) Study design – while the study design is understood in the context of the pandemic, others have used designs such as REMAP-CAP to quickly examine therapeutic options. I suggest the limitations around the open label observational nature and matching technique should be strengthened. On referring to the ClinicalTrials.gov registration NCT04357366 it seems this was posted 4 days after the first patient was enrolled with a planned recruitment of 500 patients not the 260 you have quoted in the statistics section. This disparity draws concerns and should be addressed.

3) Primary endpoint – your definition of SRF is not standard and the choice of D14 not standard either. Typically phase 2 studies would use something like ventilator-free days at D28 noting that mortality outcomes in this cohort are now reported at D90 (sometimes D60). While I understand this cannot be changed now it does require discussion and probably some consideration of the criteria for intervention (both NIV or intubation), the use of blood gases vs. pulse oximetry, and assignment of inspired oxygen to various forms of oxygen therapy.

4) Secondary endpoints – as mentioned outcome data such as mortality are now usually reported at longer endpoints than D30. Supplementary Figures 3 and 4 show similar outcomes at longer timepoints and ICU discharge is again not a standard endpoint. The symptom score is not familiar to this reviewer and needs referencing re its validity.

5) APACHE II score – Table 1 – the scores in both groups seem quite low for patients with COVID-19 requiring hospital admission (typical ward patient with pancreatitis would be 11 and average ICU admission would be 15 and 20+ for acute hypoxemic respiratory failure). The data are expressed with 2 decimal points but is captured as an integer suggesting that it should be reported to the same level of precision.

6) SOFA score – data such as the Δ SOFA score are usually regarded as problematic – I suggest just use the SOFA score data.

7) Cytokine level – in the standard care alone (matched patients) the D7 data are an n less than 20% of the original cohort which is well maintained in the anakinra group. I suggest this makes any comparisons very exploratory at best.

8) Dexamethasone effect (Appendix 1—table 2) – the data in this table need to be very carefully discussed or represented. For example it might be interpreted that those patients who developed SRF pts were protected by Anakinra, but the data show an opposite effect for dexamethasone which most would suggest has been shown to be effective in appropriately designed trials.

9) Please consider the role of IL-1α release as part of why anakinra was so successful. Also, please explain the processing of the IL-1β precursor by activation of NLRP3. It may also be worth stating that NLRP3 activation occurs early in the infection.

10) The main weakness of this study is that it is a single arm non-randomised trial. Those receiving the intervention are recruited from different hospitals than those receiving SOC. It is not clear how the SOC patients are recruited to the trial so there are concerns that immortal time bias could induce artefactual differences. The definition of the primary outcome is still not clearly presented in the Abstract.

11) Propensity matching methods are not a panacea so more detail on how this was done and even some sensitivity analysis would be warranted. Why did they not simply use all SOC patients as the comparator?

---

## [Author Response]

Essential Revisions:1) Cytokine storm – there is concern particularly from Carolyn Calfee and Michael Matthay that COVID-19 patients with ARDS do not have higher median cytokine levels than other patients with ARDS or septic shock (Sinha, Matthay and Calfee, 2020 and Lancet Resp Med Dec 2020 https://doi.org/10.1016/S2213-2600(20)30404-5). Given that various well-designed double blind trials have not shown benefit from this approach despite likely greater cytokine levels than your cohort I suggest this argument be rewritten and great care used when referring to a cytokine storm.

We thank the reviewers for bringing this up. It is true that the circulating concentrations of inflammatory cytokines are in general lower in COVID19 than in severe sepsis, but these mediators are still strongly increased, as a sign of hyperinflammation. We changed this discussion to be more nuanced (Introduction), and we added the respective reference at the beginning of the manuscript. Enumeration of the remaining references was changed accordingly.

2) Study design – while the study design is understood in the context of the pandemic, others have used designs such as REMAP-CAP to quickly examine therapeutic options. I suggest the limitations around the open label observational nature and matching technique should be strengthened.

We thank the reviewer for these comments. The limitations of the study are now better provided a):

“The lack of randomized design is acknowledged as a limitation in study design. The non-randomized design led to two more limitations: the use of SOC parallel comparators and the lack of availability of follow-up samplings on day 7”; and b):

“Several meetings held between the investigators addressed the need to adapt one placebo comparator arm of treatment; however, the available data about the efficacy of anakinra did not lead to take the decision of integration of placebo comparators.”

On referring to the ClinicalTrials.gov registration NCT04357366 it seems this was posted 4 days after the first patient was enrolled with a planned recruitment of 500 patients not the 260 you have quoted in the statistics section. This disparity draws concerns and should be addressed.

The reviewer is correct in this. However, since we are under the European Medicines Agency law, we were obliged to register the study in the public repository EudraCT before submission for approval to our National Competent Authority. This is clarified in the revised manuscript:

“Registration at the EU repository EudraCT was done on March 31^st^ 2020 before submission for regulatory approval.”

The interim analysis was pre-planned and this is now stated:

“Although the study is on-going, we present here the results of the pre-planned interim analysis according to the amendment of the SAVE trial on October 15^th^ 2020 to the National Organization for Medicines of Greece.”

3) Primary endpoint – your definition of SRF is not standard and the choice of D14 not standard either. Typically phase 2 studies would use something like ventilator-free days at D28 noting that mortality outcomes in this cohort are now reported at D90 (sometimes D60). While I understand this cannot be changed now it does require discussion and probably some consideration of the criteria for intervention (both NIV or intubation), the use of blood gases vs. pulse oximetry, and assignment of inspired oxygen to various forms of oxygen therapy.4) Secondary endpoints – as mentioned outcome data such as mortality are now usually reported at longer endpoints than D30. Supplementary Figures 3 and 4 show similar outcomes at longer timepoints and ICU discharge is again not a standard endpoint. The symptom score is not familiar to this reviewer and needs referencing re its validity.

We thank the reviewer for this comment. To better address this, ventilator-free days and mortality by day 90 have been added as exploratory endpoints in the revised Table 2. We agree that times until ICU discharge and until hospital discharge need to be removed. Instead, we have provided the WHO clinical progression scale by day 28 as exploratory endpoint in the newly revised Figure 4. One more figure, namely Appendix 1—figure 2 has been added with survival until day 90. Three references on the score of respiratory symptoms, namely Ramirez et al., 2019; Stets et al., 2019 and Barrera et al., 2016, have been added.

5) APACHE II score – Table 1 – the scores in both groups seem quite low for patients with COVID-19 requiring hospital admission (typical ward patient with pancreatitis would be 11 and average ICU admission would be 15 and 20+ for acute hypoxemic respiratory failure). The data are expressed with 2 decimal points but is captured as an integer suggesting that it should be reported to the same level of precision.

We do agree with the reviewer that APACHE II score is higher when patients with acute hypoxemic failure are admitted to hospital. However, we need to emphasize that participants in the SAVE trial and their SOC parallel comparators were not started in the study because of acute hypoxemic failure, but because of pneumonia requiring hospitalization. Treatment with anakinra was started in order to avoid progression into acute hypoxemic failure and this is the reason why a respiratory ratio less than 150 mmHg was a criterion of exclusion from the study. With this in mind the low APACHE II scores are expected. The expression of data in Table 1 was revised, as suggested.

6) SOFA score – data such as the Δ SOFA score are usually regarded as problematic – I suggest just use the SOFA score data.

This is now provided in the revised Table 2, as suggested.

7) Cytokine level – in the standard care alone (matched patients) the D7 data are an n less than 20% of the original cohort which is well maintained in the anakinra group. I suggest this makes any comparisons very exploratory at best.

This limitation is now addressed in the Discussion section of the revised manuscript.

8) Dexamethasone effect (Appendix 1—table 2) – the data in this table need to be very carefully discussed or represented. For example it might be interpreted that those patients who developed SRF pts were protected by Anakinra, but the data show an opposite effect for dexamethasone which most would suggest has been shown to be effective in appropriately designed trials.

The manuscript now reads:

“The reported higher frequency of dexamethasone intake among patients who developed SRF should not be interpreted as causality; it does simply reflect that the prescription of dexamethasone was greater among patients who were considered more severe by the treating physicians.”

9) Please consider the role of IL-1α release as part of why anakinra was so successful. Also, please explain the processing of the IL-1β precursor by activation of NLRP3. It may also be worth stating that NLRP3 activation occurs early in the infection.

These are very relevant aspects raised by the reviewers. The manuscript now reads:

“Recent data suggest early activation of the NLRP3 inflammasome in severe COVID-19; the formation of the end product caspase-1 was enhanced among patients who had unfavourable outcome. […] Anakinra also inhibits IL-1α. SARS-CoV-2 infection is suggested to cause massive release of IL-1α followed by sensing from monocytes and tissue macrophages and further activation of the NLRP3 inflammasome leading to perpetuation of the pro-inflammatory responses (Cavalli et al., 2021).” Three new references, namely Rodrigues et al., 2021; Dinarello et al., 1987 and Cavalli et al., 2021, were added.

10) The main weakness of this study is that it is a single arm non-randomised trial. Those receiving the intervention are recruited from different hospitals than those receiving SOC. It is not clear how the SOC patients are recruited to the trial so there are concerns that immortal time bias could induce artefactual differences.

We thank the reviewer for these challenging comments. The limitations of the study are now better provided a):

“The lack of randomized design is acknowledged as a limitation in study design. The non-randomized design led to two more limitations: the use of SOC parallel comparators and the lack of availability of follow-up samplings on day 7”; and b):

“Several meetings held between the investigators addressed the need to adapt one placebo comparator arm of treatment; however, the available data about the efficacy of anakinra did not lead to take the decision of integration of placebo comparators.”

The selection procedure of the SOC parallel comparators is now fully described, reading:

“The procedure to identify appropriate comparators treated in parallel with SOC was as follows: a) patients receiving SOC were hospitalized in seven medical departments that participate in the HSSG (https://www.sepsis.gr/). […] Clinical data of these patients on the day of hospital admission and start of SOC were used for comparison to the baseline data of participants of the SAVE trial.”

The definition of the primary outcome is still not clearly presented in the Abstract.

This was added in the Abstract, as suggested.

11) Propensity matching methods are not a panacea so more detail on how this was done and even some sensitivity analysis would be warranted. Why did they not simply use all SOC patients as the comparator?

We thank the review for this comment. We have now provided the information for the total available parallel SOC comparators in Table 1. As you may see, their baseline severity was higher than anakinra-treated patients, as far as APACHE II, SOFA score and pneumonia severity index are concerned. This generated the need for propensity score matching. The comparative time to progression into severe respiratory failure for all SOC comparators and the 130 anakinra-treated patients is now provided in Figure 1—figure supplement 1. The manuscript now reads:

a) “The inclusion of 179 parallel SOC comparators was done within the same time frame. […] To match for these differences, propensity score-matching was done and 130 fully-matched parallel SOC comparators were selected (Table 1);” and

b) “The superiority of anakinra treatment was also documented over the total of 179 available SOC parallel comparators (Figure 1—figure supplement 1). However, the baseline differences between the total SOC parallel comparators and the participants of the SAVE trial, led to limit the remaining analysis between the 130 anakinra-treated participants of the SAVE trial and the 130 fully matched SOC parallel comparators.”